# Sella Turcica and Cranial Base Symmetry in Anterior Synostotic Plagiocephaly Patients: A Retrospective Case–Control Study

**DOI:** 10.3390/diagnostics15172199

**Published:** 2025-08-29

**Authors:** Edoardo Staderini, Davide Guerrieri, Michele Tepedino, Gianmarco Saponaro, Alessandro Moro, Giulio Gasparini, Patrizia Gallenzi, Massimo Cordaro

**Affiliations:** 1Postgraduate School of Orthodontics, Catholic University of the Sacred Heart, 00168 Rome, Italy; edoardo.staderini@unicatt.it (E.S.); patrizia.gallenzi@unicatt.it (P.G.); massimo.cordaro@unicatt.it (M.C.); 2School of Dentistry, Catholic University of the Sacred Heart, 00168 Rome, Italy; davideguerrieri99@gmail.com; 3Department of Biotechnological and Applied Clinical Sciences, University of L’Aquila, 67100 L’Aquila, Italy; 4Maxillo-Facial Surgery, Unit Catholic University of the Sacred Heart, 00168 Rome, Italy; gianmarco.saponaro@unicatt.it (G.S.); alessandro.moro@unicatt.it (A.M.); giulio.gasparini@unicatt.it (G.G.)

**Keywords:** plagiocephaly, sella turcica, skull base

## Abstract

**Background/Objectives**: The present case–control study aims to compare the symmetry of the sella turcica and cranial base of nine patients with anterior unicoronal synostotic plagiocephaly (ASP) and nine healthy patients referred to the maxillofacial unit of the Fondazione Policlinico Universitario Agostino Gemelli. The primary aim of this study is to assess changes in the morphology of the sella turcica and skull base in comparison with a healthy control population using both a 2D and 3D analysis of the sella turcica and skull base. **Methods**: Computed tomography (CT) scans of nine ASP patients from the Fondazione Policlinico Universitario Agostino Gemelli in Rome were retrieved. A quantitative evaluation of the skull base and the sella turcica was performed through the asymmetry index (A.I.), obtained from the comparison of the point-to-point distances ipsilateral and contralateral to the synostosis. A qualitative three-dimensional (3D) evaluation of the asymmetry of the sella turcica was performed by comparing each sella model with its mirrored counterpart; then, the root mean square (RMS) displacement between the original and mirrored 3D models was calculated. **Results**: The results showed higher A.I. values in the study group, particularly the length of the anterior cranial fossa, with A.I. values of 7.96 (study) vs. 0.02 (control). **Conclusions**: The higher values of the asymmetry index observed in the study group supported the presence of statistically significant asymmetries in the sella and cranial fossa measurements compared to the control group.

## 1. Introduction

The term “plagiocephaly” derives from the Greek words “plagio” (oblique) and “kephale” (head), thus describing an asymmetric skull [1].

Anterior synostotic plagiocephaly refers to craniosynostosis of the hemicoronal suture and is the third most common form of craniosynostosis, with an incidence of 13–16% among all and affecting 1 in every 1000 births. The right side is more affected than the left, with a male-to-female ratio of 1:2; 61% of cases are sporadic and 39% are syndromic [2,3].

The pathognomonic features of anterior synostotic plagiocephaly (ASP) are unilateral flattening of the forehead and shallow orbit. Ocular protrusion, strabismus, torticollis, deviation of the nasal pyramid and mandible, and retraction of the malar prominence may occur. The ear and glenoid fossa of the affected side are advanced, with anterior displacement of the temporomandibular joint. The nasal pyramid and chin symphysis are deviated towards the opposite side, giving a scoliotic appearance to the face [2,4,5,6].

The initial growth of the cranial base precedes that of other regions, and the amount of growth at the spheno-occipital synchondrosis is relatively limited. Therefore, the cranial base remains relatively stable from early infancy, especially when compared to the facial region [7,8,9]. The sphenoid bone completes its configuration and development in the early stages of life, and it is believed to maintain stability during cranial growth due to the presence of foramina, through which nerves and blood vessels pass. Furthermore, the sphenoid bone retains a constant shape during the growth period due to its cartilaginous growth pattern and growth regulation mediated by intrinsic genetic factors [10,11]. Considering these characteristics, the sphenoid bone serves as a reference point in the analysis of craniofacial asymmetry, and the cranial base is defined as a reference plane for craniofacial region measurements [7,10].

Moreover, some studies have shown that the sella turcica complex exhibits reduced interindividual variability between affected and unaffected patients with plagiocephaly [12,13].

The main methods for assessing facial asymmetry are based on the identification of anthropometric landmarks on three-dimensional images (point-based method) or considering specific “areas of symmetry/asymmetry” of the face (surface-based method) [14,15].

Landmark-based and surface-based methods provide distinct approaches for assessing facial asymmetry. Landmark-based methods quantify asymmetry on a limited set of predefined points on the face. In addition, asymmetry of facial regions where landmarks are not placed may easily be underdiagnosed. In contrast, surface-based methods use a dense point cloud to capture the entire facial surface, offering a more comprehensive evaluation. However, an overall facial assessment might not be helpful in detailing clinically significant asymmetry in specific facial areas [14,15]. The present study employed both abovementioned analyses.

The primary aim of this study was to assess changes in the morphology of the sella turcica and skull base in comparison with a healthy control population using both a 2D and 3D analysis of the sella turcica and skull base.

## 2. Materials and Methods

### 2.1. Study Design

The present single-center retrospective case–control study was carried out in the department of Orthodontics of the “Fondazione Policlinico Universitario A. Gemelli IRCCS (Rome)” teaching hospital. The study protocol was approved by the Internal Review Board of Fondazione Policlinico Universitario A. Gemelli IRCCS (ID: 6282, date of approval: 11 January 2024. All the procedures were in accordance with the Helsinki Declaration of 1975 and its further amendments. Prior to the analysis, a researcher contacted the caregivers to explain the study protocol and obtain written informed consent and privacy policies. This study was reported according to the STROBE guidelines Appendix A).

### 2.2. Sample Size Calculation

Based on the study by Calandrelli et al. [16], the post hoc sample size calculation indicated a total of 18 patients, 9 per group, to obtain statistically significant differences by setting a first species alpha error of 5%, a beta power of 85%, and an effect size of 1.59, with a two-tailed Student’s *t*-test.

Therefore, a total of 18 patients were retrospectively enrolled, 9 of whom had plagiocephaly and 9 of whom were healthy without plagiocephaly.

### 2.3. Patient Selection

This study was performed on fully anonymized computed tomography (CT) scans of 9 patients with anterior unicoronal synostotic plagiocephaly as the study group and 9 healthy patients without this condition as the control group. The CT scans were retrieved from the hospital database; the enrolled patients were referred to the Maxillofacial Unit of the Fondazione Policlinico Universitario Agostino Gemelli IRCCS from 2010 to 2011. Regarding the study group, the inclusion criteria were as follows:-Patients with anterior synostotic plagiocephaly;-Adult patients of Caucasian ethnicity between 20 and 40 years of age;-Patients with CT examinations of the facial massif.

The exclusion criteria of the study group were as follows:-Patients with a history of orthodontic treatment or orthognathic surgery;-Patients with a history of trauma and neoplasms in the craniofacial region;-Patients with a long-term history of drug use that may have affected bone development.

### 2.4. Protocol and Measurements for 2D Point-Based Quantitative Analysis

The analysis was performed on CT scans of the facial massif exported as Digital Imaging and Communications of Medicine (DICOM) files. Subsequently, the three-dimensional images were imported into Mimics Medical software (version 19.0, Materialise, Leuven, Belgium).

The following procedure was performed:-CT orientation: The CT scans were reoriented using the axial, coronal, and sagittal reference planes;-CT segmentation: The following software tools were used in the following order: “Thresholding” was used to identify a “mask” with the cranial base through manual selection of pixel intensity values; “Crop mask” allowed for outlining the region of interest; and “Calculate 3D” enabled the generation of a three-dimensional model from the contours of the previously segmented mask.

### 2.5. Two-Dimensional Point-Based Quantitative Analysis: Sella Turcica

The morphology of the sella turcica was analyzed according to the protocol proposed by Ortiz and Ugurlu [17,18]. The following anatomical landmarks were identified on the sella turcica (Figure 1) (Table 1) [19].

The following linear measurements of the sella turcica were calculated based on the anatomical landmarks of reference [19]:-Anterior clinoid distance: The distance between the right and left anterior clinoid processes (ACPs).-Posterior clinoid distance: The distance between the right and left posterior clinoid processes (PCPs).-Left sella length: The distance between the left (L) anterior and left posterior clinoid processes.-Right sella length: The distance between the right (R) anterior and right posterior clinoid processes.-Median sella length: The distance between the tuberculum sellae (TS) and the dorsum sellae (DS).

### 2.6. Two-Dimensional Point-Based Quantitative Analysis: Skull Base

The skull base was split into two hemibases by an anatomical midline that runs from the anterior edge of the crista galli (C) to the sella turcica (S) and then to the opisthion (O). If the CSO angle was approximately 180 degrees, it meant that the two hemibases were almost symmetrical. The CSO angle was divided into three parts:-The CSX angle for the anterior cranial fossa, between the crista galli (C), sella turcica (S), and xiphoid process of the lesser wing of the sphenoid (X);-The XSM angle for the middle cranial fossa, between the xiphoid process (X), sella turcica (S), and internal acoustic meatus (M);-The MSO angle for the posterior cranial fossa, between the internal acoustic meatus (M), sella turcica (S), and opisthion (O).

The landmarks identified on the skull base for the lengths of the anterior, middle, and posterior cranial fossae were the anterior edge of the crista galli (C), the xiphoid process of the lesser wing of the sphenoid (X), the internal acoustic meatus (M), and the opisthion (O). The lengths between C, X, M, and O were measured on both sides:-CX length for the anterior cranial fossa between the crista galli (C) and xiphoid process of the lesser wing of the sphenoid (X);-XM length for the middle cranial fossa between the xiphoid process (X) and the internal acoustic meatus (M);-MO length for the posterior cranial fossa between the internal acoustic meatus (M) and opisthion (O).

The asymmetry of the skull was assessed by comparing the differences in angles and lengths between the two sides [16,20]. (Figure 2)

### 2.7. Qualitative 3D Surface-Based Analysis: Sella Turcica

The three-dimensional images generated with Mimics were subsequently imported into 3-Matic Medical software (version 11.0, Materialise, Leuven, Belgium). The sella turcica contour was mirrored with a three-step process:-Firstly, the “new sketch” tool was used to generate a cutting plane passing by a selection of three median landmarks (Figure 3): the tuberculum sellae (TS), the sella turcica floor (STF) (lowest point of the pituitary fossa of the sphenoid), and the dorsum sellae (DS);-Secondly, the “Cut” function was performed to split the sella turcica along the abovementioned plane into two hemi-portions: the “unhealthy side” (ipsilateral to the ASP) from the “healthy side” (contralateral to the ASP);-Thirdly, the “Mirror” function was used to create a mirror copy of the “unhealthy side”.

A similar mirroring process was applied to the control group, where the left side of the sella turcica was chosen as the arbitrary side to be superimposed.

The STL files derived were subsequently imported into Geomagic Control 2014 software (3D Systems, Morrisville, NC, USA).

The STLs of the “unhealthy side” (selected as “reference”) underwent point-based superimposition into the “healthy side” (selected as “test”) using three median landmarks: the tuberculum sellae (TS), the sella turcica floor (STF), and the dorsum sellae (DS).

Following the point alignment process, the “best fit” and then the “3D comparison” functions were used, which the software uses to compare the surface of the “reference” model with that of the “test” model, calculating the differences point by point. The RMS was used to assess the three-dimensional displacement between the “test” and “reference”. A low RMS value indicates a high similarity between the “test” and “reference”.

This generates a color map that visually shows the differences (Figure 4):

Green: The test model is within the tolerance limits.

Red: Outward displacement of the test model compared to the reference.

Blue: Inward displacement of test model material compared to the reference.

### 2.8. Statistical Analysis

All measurements and visual assessments were independently reviewed and confirmed by two experienced clinicians (D.G. and E.S.), and the average between those measurements was used. The operators performed all measurements twice, within at least a 2-week interval, using identical datasets, equipment, and protocols. Intra- and inter-operator reliabilities were evaluated using Dahlberg’s formula (s = √Σd^2^/2n, where “d” represents the difference between repeated measurements) [21].

The asymmetry index (A.I.) formula was applied to calculate the asymmetry index for linear and angular measurements of the sella turcica and skull base [22].

In the control group, lacking a healthy side and a diseased side as patients without the medical condition, the right side was arbitrarily chosen as the healthy side and the left side as the diseased side. In addition, the asymmetry index values were taken as absolute values. The formula for calculating the asymmetry index used for linear and angular measurements of the sella turcica and skull base.Asymmetry index (A.I.) = ((Healthy side – Affected side)/(Healthy side + Affected side)) × 100(1)

The qualitative variables measured using Geomagic Control 2014 software (3D Systems, Morrisville, NC, USA) were analyzed by the statistical measure RMS “root mean square” used to assess the accuracy of a 3D scan compared to the reference model [23]. The calculation of the RMS value for each three-dimensional image was performed by the software itself.

Statistical analysis was performed using Excel 2016 software (Microsoft, Redmond, WA, USA) to calculate the *p*-value given by Student’s *t*-test for the analysis of 2D quantitative and 3D qualitative RMS measurements. The individual values of each patient in the study group and the control group were compared in the *t*-test calculation for each single measurement.

Statistical significance was set at *p*-value < 0.05.

## 3. Results

### 3.1. Two-Dimensional Asymmetry

Using Excel 2016 software (Microsoft, Redmond, WA, USA), the mean values and standard deviations, expressed in millimeters, for each measurement for both the study group and the control group were calculated. Each numerical value was approximated to the first two decimal numbers, and the resulting values were divided into three tables associated with the *p*-value given by the *t*-test calculation, with the individual patient values referring to the single measurement (Table 2, Table 3 and Table 4).

Based on the point-based measurements, the greatest asymmetry index differences were found for the following:-Sella length (defined as the comparison between the right and left sella lengths), with asymmetry index values of 5.94 (study) vs. 1.62 (control);-The anterior cranial fossa length measurements (defined as the comparison between the right and left CX lengths) were 39.62 ± (4.89) for the ill side and 46.50 ± (5.56) for the healthy side in the study group vs. 41.98 ± (3.22) for the ill side and 41.96 ± (3.23) for the healthy side, with asymmetry index values of 7.96 (study) vs. 0.02 (control);-The anterior cranial fossa angle widths (defined as the comparison between the right and left CSX angles) were 48.23 ± (6.80) for the ill side and 58.25 ± (7.43) for the healthy side in the study group vs. 51.84 ± (2.42) for the ill side and the 52.43 ± (3.76) for healthy side, with asymmetry index values of 7.96 (study) vs. 0.02 (control) and asymmetry index values of 9.42 (study) vs. 0.51 (control).

### 3.2. Three-Dimensional Asymmetry

Using Excel 2016 software (Microsoft, Redmond, WA, USA), the standard deviation and the *p*-value given by the *t*-test were calculated.

The mean RMS values were 1.00 (±0.16) for the study group vs 0.73 (±0.16) for the control group, with *p* = 0.003

### 3.3. Reliability

The calculated error was 0.4 and 0.5 for the intra- and inter-operator reliabilities, respectively.

## 4. Discussion

### 4.1. Discussion of the Results of the Two-Dimensional Point-Based Quantitative Analysis

The results of the paired *t*-test for the two-dimensional linear and angular measurements revealed no statistically significant differences between the study group and the control group for all measures taken.

However, the asymmetry index revealed a statistically significantly higher value in the study group patients than in the healthy control group patients for all the measurements taken. The higher values of the asymmetry index observed in the study group supported the presence of statistically significant asymmetries in the sella and cranial fossa measurements compared to the control group.

The 3D superimposition confirmed the presence of macro-morphological changes in the sella turcica.

The present data suggested that unicoronal synostotic plagiocephaly revealed a lack of alignment of the midline structures associated with deviation of the facial structure towards the synostotic side. The flexion of the cranial base has a rotational pivot at the anterior clinoid process. The cranial base tends to be smaller on the synostotic side, and midline structures located anterior to the sella turcica, including those from the sella tubercle to the optic foramen, show deviations towards the same side [24,25].

The coronal suture is derived from the neural crest, and the parietal bone is derived from the mesoderm, therefore requiring precise regulation of progenitor cells for proper suture development. Embryological defects in this process, such as alteration of mesenchymal stem cell preservation at the interface between the neural crest and mesoderm, can lead to premature osteogenic differentiation. This causes early fusion of the coronal suture, disrupting the coordinated growth of the cranial vault and contributing to the asymmetrical craniofacial morphology observed in anterior synostotic plagiocephaly [26].

The early unilateral synostosis of the coronal suture causes a localized growth defect of the anterior and middle cranial fossae of the affected side with consequent expansion of the opposite anterior cranial fossa. This is the reason why the anterior cranial fossa angles and lengths showed an asymmetry between the groups, with higher values found for the healthy sides in ASP patients. The asymmetry of the posterior cranial fossa, on the other hand, is predominantly found in groups IIB and III and may be due to anterior displacement of the petrous fossa; these findings agree with previous studies that reported that the skull base has a rigid and integral biomechanical structure, deviated towards the synostotic side [16,20].

### 4.2. Discussion of the Results of the Three-Dimensional Surface-Based Qualitative Analysis

The results of the paired *t*-test showed a statistically significant difference in the RMS error between the two groups (*p* = 0.003). This result indicated that 3D morphological alterations are indeed present and detectable when analyzing the entire surface morphology of the sella.

### 4.3. Limitations of This Study

The primary limitation of the present study was the small sample size of nine patients per group. Although the study population achieved adequate statistical power according to the post hoc analysis, the findings should be regarded as preliminary and validated in larger cohort studies. In particular, it remains unclear whether the lack of statistical significance of the 2D results was attributable to the limited sample size, which, in this case, was constrained by the rarity of the ASP condition. Nevertheless, the findings suggest that the asymmetry index algorithm may represent a more sensitive approach for quantifying asymmetry than direct raw measurements, even in studies with a small patient sample.

Descriptive analysis based on a comparison of A.I.s according to gender within the study group showed that women had a higher average asymmetry index for the length of the sella turcica than men (7.84 vs. 3.57). Conversely, men showed higher asymmetry indices for the lengths of the anterior and middle cranial fossae, as well as for the angles of the anterior and middle cranial fossae. These patterns may suggest sex-related variations in skull base morphology in patients with ASP. However, due to the limited sample size, these results should be interpreted with caution. Further investigation with larger samples is needed to clarify the role of sex in influencing skull base asymmetry in this patient population.

## 5. Conclusions

Patients with anterior synostotic plagiocephaly showed statistically significant differences in the asymmetry indices for the sella turcica and cranial base compared to the control group, suggesting macro-morphological alterations of the abovementioned structures. The 3D superimposition confirmed the alteration of the morphology of the sella turcica.

In conclusion, the data findings suggested that the sella turcica and skull base were not stable in patients with anterior synostotic plagiocephaly and that, although the 2D measurements did not reach the threshold for the statistical significance in the paired *t*-test, the asymmetries detected through the asymmetry index could have a relevant clinical impact on the stability of the sella turcica and skull base, highlighting the importance of in-depth analysis and continuous monitoring in craniosynostosis to improve clinical and surgical outcomes. The usefulness of the asymmetry index lies in its ability to clearly highlight an asymmetry that a simple comparison between the two sides does not reveal as statistically significant. Therefore, the same algorithm could be applied to assess facial asymmetry in other complex skeletal malformations like scaphocephaly or trigonocephaly.

Future research should evaluate whether compensatory facial growth, such as mandibular or orbital adaptations, contributes to asymmetry of the skull base and sella turcica, in order to better understand their contribution to the morphological variations observed. These results underlined the need for further studies with larger samples to validate this study and to develop more targeted interventions for the management of anterior synostotic plagiocephaly.

## Figures and Tables

**Figure 1 diagnostics-15-02199-f001:**
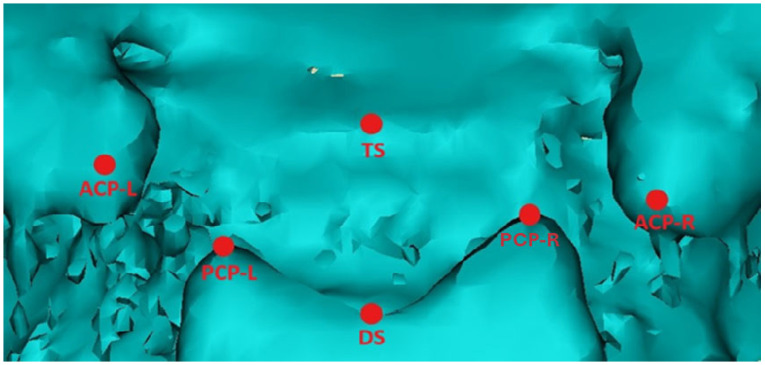
Three-dimensional image of the sella turcica of a patient with anterior synostotic plagiocephaly imported into Mimics software, with landmarks identified for the two-dimensional measurements.

**Figure 2 diagnostics-15-02199-f002:**
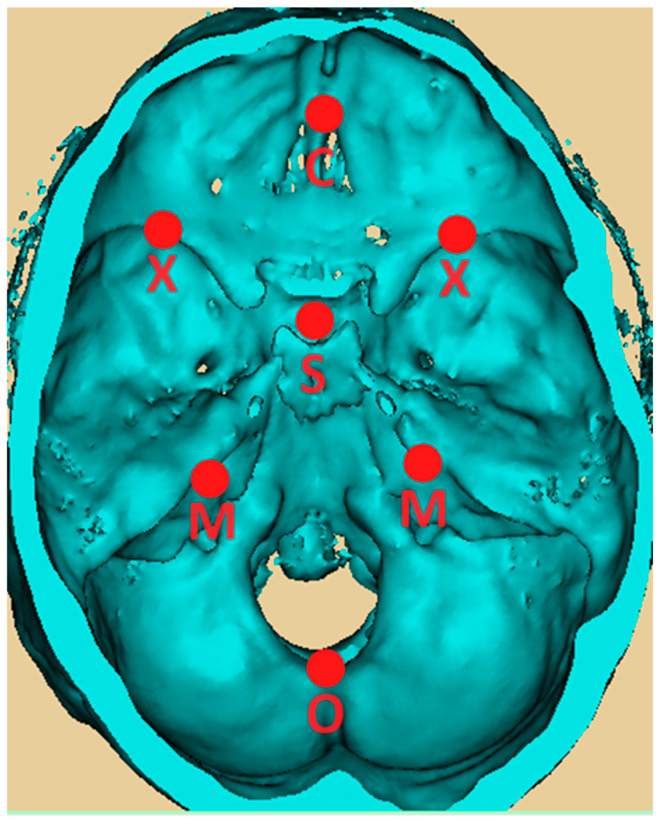
Three-dimensional image of the skull base of a patient with anterior synostotic plagiocephaly imported into Mimics software, with landmarks identified for the two-dimensional measurements.

**Figure 3 diagnostics-15-02199-f003:**
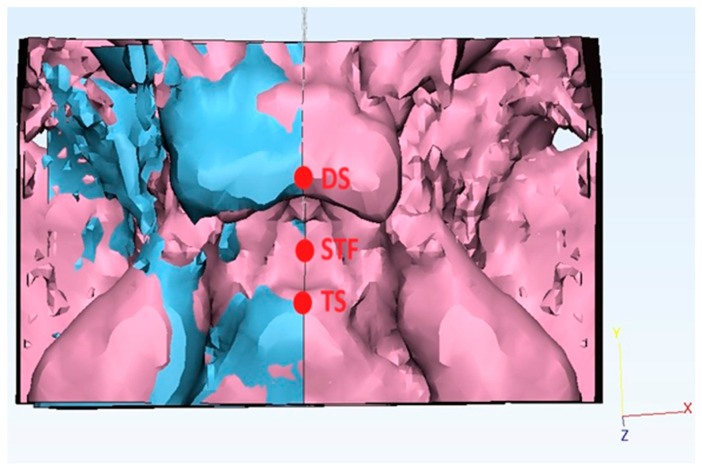
Example of a three-dimensional superimposition of the sella turcica asymmetry with 3-Matic software, with a point-based alignment of the affected side (after mirroring) on the healthy side of a patient with anterior synostotic plagiocephaly landmarks.

**Figure 4 diagnostics-15-02199-f004:**
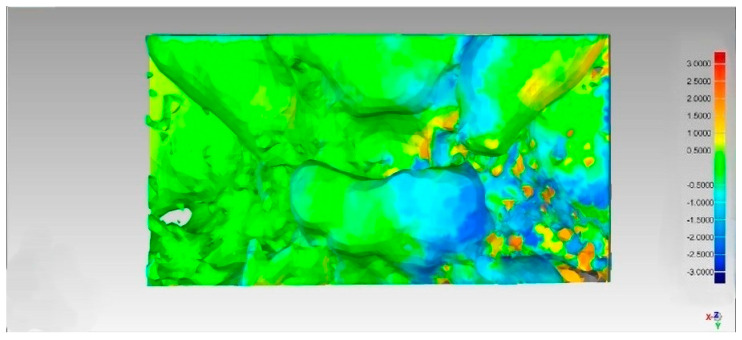
Example of a three-dimensional color map created with Geomagic software, showing the 3D displacement between the affected and healthy side of the sella turcica. Color map: Green: The test model is within the tolerance limits. Red: Outward displacement of the test model compared to the reference. Blue: Inward displacement of test model material compared to the reference.

**Table 1 diagnostics-15-02199-t001:** Description of anatomical landmarks on the sagittal and coronal planes.

Landmark	Sagittal Plane Description	Coronal Plane Description
ACP-R (anterior clinoid process—right side)	Most anterior and superior point of the right anterior clinoid process apex.	Most lateral and superior projection of the right anterior clinoid process.
ACP-L (anterior clinoid process—left side)	Most anterior and superior point of the left anterior clinoid process apex.	Most lateral and superior projection of the left anterior clinoid process.
PCP-R (posterior clinoid process—right side)	Most posterior and superior point of the right posterior clinoid process apex.	Most medial and superior projection of the right posterior clinoid process.
PCP-L (left posterior clinoid process—left side)	Most posterior and superior point of the left posterior clinoid process apex.	Most medial and superior projection of the left posterior clinoid process.
DS (Dorsum sellae point)	Most posterior and superior point on the midline of the dorsum sellae upper margin.	Central point on the upper margin of the dorsum sellae, equidistant from PCP-R and PCP-L.
TS (Tuberculum sellae)	Most anterior and superior point on the midline of the tuberculum sellae margin.	Central point on the anterior margin of the sella turcica, equidistant from ACP-R and ACP-L.

**Table 2 diagnostics-15-02199-t002:** Mean value and standard deviation of linear measurements of the sella turcica.

Linear Measurements	Study Group	Control Group	*p*-Value
Anterior Clinoid Distance	23.20 ± (2.82)	24.28 ± (2.67)	0.41
Posterior Clinoid Distance	13.88 ± (3.02)	12.13 ± (2.44)	0.19
Median sella Length	9.29 ± (1.31)	9.53 ± (1.35)	0.71
Diseased-side sella length	7.21 ± (1.68)	7.57 ± (1.83)	0.67
Healthy-side sella length	8.16 ± (1.97)	7.29 ± (1.64)	0.33
A.I. Sella length	5.94	1.62	0.02*

An asterisk is indicated for *p*-values < 0.05.

**Table 3 diagnostics-15-02199-t003:** Mean value and standard deviation of linear measurements of the skull base.

Linear Measurements	Study Group	Control Group	*p*-Value
Ill-side anterior cranial fossa length	39.62 ± (4.89)	41.98 ± (3.22)	0.24
Healthy-side anterior cranial fossa length	46.50 ± (5.56)	41.96 ± (3.23)	0.05
A.I. anterior cranial fossa length	7.96	0.02	0.01*
Ill-side median cranial fossa length	53.16 ± (3.46)	56.89 ± (6.08)	0.12
Healthy-side median cranial fossa length	56.27 ± (3.22)	56.23 ± (6.21)	0.98
A.I. middle cranial fossa length	2.86	0.62	0.00*
Ill-side posterior cranial fossa length	46.03 ± (4.39)	44.32 ± (6.37)	0.51
Healthy-side posterior cranial fossa length	45.85 ± (5.02)	44.16 ± (5.67)	0.51
A.I. posterior cranial fossa length	0.25	0.05	0.79

The terms ill side and healthy side refer to the ipsilateral and contralateral sides of plagiocephaly, respectively. In addition, an asterisk is indicated for *p*-values < 0.05.

**Table 4 diagnostics-15-02199-t004:** Mean value and standard deviation of angular widths of the skull base.

Angular Widths	Study Group	Control Group	*p*-Value
Ill-side anterior cranial fossa angle	48.23 ± (6.80)	51.84 ± (2.42)	0.15
Healthy-side anterior cranial fossa angle	58.25 ± (7.43)	52.43 ± (3.76)	0.05
A.I. anterior cranial fossa angle	9.42	0.51	0.02 *
Ill-side middle cranial fossa angle	84.27 ± (4.52)	87.82 ± (8.64)	0.29
Healthy-side middle cranial fossa angle	88.69 ± (3.87)	89.00 ± (8.79)	0.92
A.I. middle cranial fossa angle	2.57	0.65	0.09
Ill-side posterior cranial fossa angle	40.6 ± (2.57)	40.48 ± (7.32)	0.96
Healthy-side posterior cranial fossa angle	40.05 ± (3.53)	40.26 ± (6.78)	0.93
A.I. posterior cranial fossa angle	0.76	1.77	0.62

The terms ill side and healthy side refer to the ipsilateral and contralateral sides of plagiocephaly, respectively. In addition, an asterisk is indicated for *p*-values < 0.05.

## Data Availability

The original contributions presented in this study are included in the article/Appendix A. Further inquiries can be directed to the corresponding author.

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
