# Peer review of "Sella Turcica and Cranial Base Symmetry in Anterior Synostotic Plagiocephaly Patients: A Retrospective Case–Control Study"

_diagnostics, 2025, doi:10.3390/diagnostics15172199_

Round 1

Reviewer 1 Report

Comments and Suggestions for Authors

This report quantitatively evaluates cranial base morphology using reconstructed 2D and 3D computerized tomography images. As a representative case, it focuses on patients with anterior synostotic plagiocephaly. The results demonstrated that the asymmetry index (A.I.), an original metric devised by the authors, was significantly higher in the patient group than in the control group. Rather than simply comparing numerical values, the use of the A.I. was both innovative and insightful. This study serves as a foundational report for research on cranial base development and offers valuable insights for pediatric neurosurgeons and related specialists.

  1. Introduction:
    The introduction is clear and detailed; no issues are noted.
  2. Materials:
    Subjects were selected appropriately based on ethical considerations.
  3. Methods:
  1. It is critically important to describe whether the measurements were taken multiple times (at least three) by multiple observers (at least three), and whether the average of those measurements was used.
  2. The selection of anatomical landmarks (ACP, PCP, TS, DS, STF) must be presented objectively and clearly.
  1. Results:
    No problems are found with the statistical analysis.
  2. Discussion:
    There are no issues with the interpretation of the results.
  1. As a reviewer’s request, please consider relating the findings to embryology and developmental biology in the discussion.
  2. As a future direction, it would be valuable to apply the same method to other conditions and other anatomical regions from an embryological perspective. We look forward to the authors’ continued research in this area.
Comments on the Quality of English Language

I can't decide comment, because I am not native English speaker.

Author Response

Reviewer’s comment:

This report quantitatively evaluates cranial base morphology using reconstructed 2D and 3D computerized tomography images. As a representative case, it focuses on patients with anterior synostotic plagiocephaly. The results demonstrated that the asymmetry index (A.I.), an original metric devised by the authors, was significantly higher in the patient group than in the control group. Rather than simply comparing numerical values, the use of the A.I. was both innovative and insightful. This study serves as a foundational report for research on cranial base development and offers valuable insights for pediatric neurosurgeons and related specialists.

Response to the Reviewer:

We thank the reviewer for the positive evaluation and encouraging feedback. No changes were requested.

Reviewer’s comment:

  1. Introduction:

The introduction is clear and detailed; no issues are noted.

  1. Materials:

Subjects were selected appropriately based on ethical considerations.

Response to the Reviewer:

Thanks for the positive feedback. No issues are noted.

Reviewer’s comment:

  1. Methods:

It is critically important to describe whether the measurements were taken multiple times (at least three) by multiple observers (at least three), and whether the average of those measurements was used.

Response to the Reviewer:

We thank the reviewer for taking this aspect up. To acknowledge this point in the present study, the following sentence has been added in the revised manuscript: “All measurements and visual assessments were independently reviewed and confirmed by two experienced clinicians (D.G. and E.S.), and the average between those measurements was used”. Please see page 8 (lines 213-218).

Reviewer’s comment:

The selection of anatomical landmarks (ACP, PCP, TS, DS, STF) must be presented objectively and clearly.

Response to the Reviewer:

We thank the reviewer for taking this aspect up. The selection of anatomical landmarks has been clarified in the revised manuscript. Please see Table 1, page 4.

Reviewer’s comment:

Results:

No problems are found with the statistical analysis

Discussion:

There are no issues with the interpretation of the results.

Response to the Reviewer:

Thanks for the positive feedback. No issues are noted.

Reviewer’s comment:

As a reviewer’s request, please consider relating the findings to embryology and developmental biology in the discussion.

Response to the Reviewer:

Thank you for the kind suggestion. As suggested, we have generalized the results in the context of embryology and developmental biology in the discussion. Please refer to page 11-12 (lines 294-300).

Reviewer’s comment:

As a future direction, it would be valuable to apply the same method to other conditions and other anatomical regions from an embryological perspective. We look forward to the authors’ continued research in this area

Response to the Reviewer:

Thank you for your observation. Accordingly, we have added a sentence with implication for further research in the revised manuscript. Please refer to pages 12 (lines 334-336).

Reviewer 2 Report

Comments and Suggestions for Authors

Dear Authors,
Thank you very much for the opportunity to review the results of your research on Sella Turcica and Cranial Base Symmetry in Anterior Synostotic Plagiocephaly Patients: A Retrospective Case-Control Study. This is a discussion of very interesting material on the possibility of assessing facial asymmetry in plagiocephaly. The article is written in a clear and easy-to-understand manner, maintaining a cause-and-effect relationship. The statistical analysis is conducted appropriately.
As a reviewer, I kindly request that the methods subsection in the abstract be shortened. A detailed discussion of asymmetry can be provided in the materials and methods section of the main manuscript. The abstract should provide an outline of the results discussed later and serve as an introduction to the manuscript.

The methodology was not discussed in the description of the statistical methods used to calculate the sample size.
Due to the significant limitation discussed in the limitations section, I suggest a more critical assessment of the obtained results, particularly in light of the small sample size. A weakness of the paper is the authors' lack of commentary on the comparison of results by gender, which would confirm or refute the literature data. In the results section, I propose a synthesis of the final conclusions. Throughout the manuscript, please correct the spelling of the significance level of p<0.001, which should be italicized.

Author Response

Reviewer’s comment:

Thank you very much for the opportunity to review the results of your research on Sella Turcica and Cranial Base Symmetry in Anterior Synostotic Plagiocephaly Patients: A Retrospective Case-Control Study. This is a discussion of very interesting material on the possibility of assessing facial asymmetry in plagiocephaly. The article is written in a clear and easy-to-understand manner, maintaining a cause-and-effect relationship. The statistical analysis is conducted appropriately.

Response to the Reviewer:

We thank the reviewer for appreciating the paper. No changes were requested.

Reviewer’s comment:

As a reviewer, I kindly request that the methods subsection in the abstract be shortened. A detailed discussion of asymmetry can be provided in the materials and methods section of the main manuscript. The abstract should provide an outline of the results discussed later and serve as an introduction to the manuscript.

Response to the Reviewer:

We thank the reviewer for the suggestion. As suggested, we have tried to summarize the abstract. Please, let us know if it is fine. Please refer to Page 1, lines 23-30.

Reviewer’s comment:

The methodology was not discussed in the description of the statistical methods used to calculate the sample size.

Response to the Reviewer:

Point taken. The calculation of the sample size has been discussed in a dedicated section. Please refer to section 2.2 (Page 3, line 92-98).

Reviewer’s comment:

Due to the significant limitation discussed in the limitations section, I suggest a more critical assessment of the obtained results, particularly in light of the small sample size.

Response to the Reviewer:

We appreciate the reviewer’s comment and fully agree with the observation. As suggested, we have revised the limitations section, emphasizing the influence of the small sample size in the interpretation of the results. Please refer to page 12 (lines 315-323).

Reviewer’s comment:

A weakness of the paper is the authors' lack of commentary on the comparison of results by gender, which would confirm or refute the literature data. In the results section, I propose a synthesis of the final conclusions.

Response to the Reviewer:

We appreciate the interesting suggestion. However, the small sample size did not allow us any subgroup analysis. This aspect was added as a limitation in the revised manuscript Please refer to page 12 (lines 324-331).

Reviewer’s comment:

Throughout the manuscript, please correct the spelling of the significance level of p<0.001, which should be italicized.

Response to the Reviewer:

Point taken. The typo has been corrected in the revised manuscript.

Reviewer 3 Report

Comments and Suggestions for Authors

ASP is a rare but significant craniofacial condition. Studying sella turcica and cranial base asymmetry may have implications for diagnosis, surgical planning, and postoperative evaluation.

Regarding this paper, I have few comments:

The sample size of 9 patients per group is quite limited. While the authors claim adequate power based on post-hoc analysis, the generalizability remains questionable.

The manuscript acknowledges that most 2D direct measurements did not reach statistical significance, yet emphasizes the A.I. differences. Clarify this discrepancy. Consider explaining whether the A.I. is more sensitive than direct raw measurements and whether it could be a more appropriate metric in small sample studies.

Did you assess intra- and inter-observer reliability for landmark identification?

Could asymmetry also be influenced by compensatory facial growth (e.g., mandible or orbit), and how did you control for this?

Author Response

Reviewer’s comment:

ASP is a rare but significant craniofacial condition. Studying sella turcica and cranial base asymmetry may have implications for diagnosis, surgical planning, and postoperative evaluation.

Response to the Reviewer:

We sincerely thank the reviewer for appreciating the clinical implication of the assessment of sella turcica and cranial base asymmetry. No changes were requested.

Reviewer’s comment:

Regarding this paper, I have few comments:

The sample size of 9 patients per group is quite limited. While the authors claim adequate power based on post-hoc analysis, the generalizability remains questionable.

Response to the Reviewer:

Thank you for the kind suggestion; the small sample size have been discussed more extensively in the revised paper. Please refer to page 12 (lines 315-323).

Reviewer’s comment:

The manuscript acknowledges that most 2D direct measurements did not reach statistical significance, yet emphasizes the A.I. differences. Clarify this discrepancy. Consider explaining whether the A.I. is more sensitive than direct raw measurements and whether it could be a more appropriate metric in small sample studies.

Response to the Reviewer:

Thanks for the valuable suggestion. The usefulness of A.I has been discussed more extensively in the revised paper. Please refer to page 12 (lines 321-323) and page 13 (lines 349-356).

Reviewer’s comment:

Did you assess intra- and inter-observer reliability for landmark identification?

Response to the Reviewer:

Thanks for the kind suggestion. Intra- and inter-observer reliability have been added in the revised manuscript. Please refer to page 8 (lines 213-218) and page 11 line 275.

Reviewer’s comment:

Could asymmetry also be influenced by compensatory facial growth (e.g., mandible or orbit), and how did you control for this?

Response to the Reviewer:

We thank the reviewer for the comment. We have added this valuable observation as interesting implication for further research, as we agree it is an important aspect. We will certainly address this point in one of our future studies. Please refer to page 13 (lines 354